# Method Development and Validation for Omega-3 Fatty Acids (DHA and EPA) in Fish Using Gas Chromatography with Flame Ionization Detection (GC-FID)

**DOI:** 10.3390/molecules26216592

**Published:** 2021-10-30

**Authors:** Suryati Muhammad Alinafiah, Azrina Azlan, Amin Ismail, Nor-Khaizura Mahmud Ab Rashid

**Affiliations:** 1Department of Nutrition, Faculty of Medicine and Health Sciences, Universiti Putra Malaysia, Serdang 43400, Selangor, Malaysia; suryati@upm.edu.my (S.M.A.); aminis@upm.edu.my (A.I.); 2Research Centre of Excellence for Nutrition and Non-Communicable Diseases, Faculty of Medicine and Health Sciences, Universiti Putra Malaysia, Serdang 43400, Selangor, Malaysia; 3Halal Products Research Institute, Universiti Putra Malaysia, Serdang 43400, Selangor, Malaysia; 4Department of Food Science, Faculty of Food Science and Technology, Universiti Putra Malaysia, Serdang 43400, Selangor, Malaysia; norkhaizura@upm.edu.my

**Keywords:** validation methods, omega-3 fatty acid, gas chromatography

## Abstract

Gas chromatography with flame ionization detection (GC-FID) has often been used to quantify fatty acids in fish. This study validated the common method for determining omega-3 fatty acids (DHA and EPA) in the raw and cooked warm-water fish, selayang, using GC-FID for subsequent evaluation on EPA and DHA retention using the Weibull model. The EPA and DHA were separated using a high-polarity capillary GC HP-88 column (60 m length, 0.25 mm ID, 0.2 μm DF) with a total run time of 45.87 min. The method was validated in linearity, precision, accuracy, specificity and sensitivity based on ICH requirements. In addition, it was found that the method had a high recovery rate (>95%) and good precision (RSD ≤ 2%) with overall RSDs ranging below 0.001% for both omega-3 PUFA. In conclusion, this method identified and quantified fatty acids and omega-3 accurately and precisely and can be used effectively for routine FAME analysis in fish samples.

## 1. Introduction

Fatty acids (FA) are essential components of the human body as the energy source vital for healthy metabolism, acting as significant cell membranes and precursors of eicosanoid hormones [1,2]. The long-chain FA, namely omega-3 and omega-6 polyunsaturated fatty acids (PUFA), are the two most vital FA due to their multiple biological roles, such as reducing oxidative stress and cardiovascular protection. The beneficial effects of long-chain PUFA consumption are related to positive effects on human health in maintaining health and reducing disease risk [3]. The critical roles of PUFA in preventing diseases may be a great tool that helps decrease the prevalence of chronic diseases worldwide. Therefore, more unsaturated fatty acids, particularly the omega-3 fatty acids, are suggested as a strategy and should be recommended as part of a daily diet. An adequate intake of omega-3 fatty acids may prevent the onset of chronic diseases, such as cardiovascular diseases, relief from the symptoms associated with ulcerative colitis, menstrual pain and joint pain [4,5,6]. Furthermore, adequate consumption of omega-3 fatty acids may also aid in optimal cognitive performance. PUFA are highly concentrated in the brain and help in the early development of cognitive function and visual sharpness [7].

Insufficient fat intake is related to certain disorders, such as abnormality in the liver, reduced growth rates, decreased immune function, depression and skin dryness. In line with this, the Food Agriculture Organization (FAO), consistent with the 2002 WHO Expert Consultation Recommendations (2003), advise adults to consume a minimum of 10% of their diet from saturated fatty acids (SFA) and between 20–35% of energy from fat. The Dietary Guidelines for Americans (2010) recommend replacing 10% of saturated fatty acids (SFA) with monounsaturated (MUFA) and polyunsaturated fatty acids (PUFA). Besides, an average of 1750 mg per week (250 mg per day) of eicosapentaenoic acid (EPA; 20:5n-3) and docosahexaenoic acid (DHA; 22:6n-3) or 8 ounces or more seafood consumption weekly is recommended [8,9].

Nonetheless, PUFA cannot be produced by the body and must be obtained from the diet. Marine foods are highly nutritious, loaded with essential nutrients such as protein, minerals, vitamins and omega-3 fatty acids. They are also excellent dietary sources of complex mixtures of saturated fatty acids (SFA), monounsaturated fatty acids (MUFA) and polyunsaturated fatty acids (PUFA) with a variety of carbon chain lengths [10]. The fatty-acid composition in fish determined by GC-FID has been studied for many decades and is still highly relevant for laboratory routine analysis until now. The crucial functions of omega-3 fatty acids on health made them essential to be analyzed accurately and quantified precisely [3].

There are various techniques and analytical methods available for the determination of the fatty acids in oil and foods worldwide. Infrared spectroscopy, capillary electrophoresis, high-performance liquid chromatography (HPLC) and gas chromatography (GC) have been used for more than a decade to analyze fatty acids [2,11]. Meanwhile, GC has been used for more than half a century to analyze fatty acids. It is an accurate, sensitive, reproducible and versatile tool for the complex mixtures that led to specific and rapid analysis of fatty acids [10,11,12,13]. While numerous analytical techniques have been implemented throughout the last decade, GC-FID remains the most appropriate, sufficient and frequently used technique to identify and quantify fatty acids in essential oils and foods [14,15,16,17]. 

Currently, the wide range of available techniques and analytical methods used in laboratories urges the need to define the scope and quality of the information obtained from them regarding internationally agreed standards [12]. To ensure the analytical approach used for a specific test is adequate for its intended use, method validation and development are required [12]. Reliable analytical data are essential for accurate interpretation of chromatography analysis for the evaluation of scientific studies and routine work. A correct interpretation of findings is critical to avoid false interpretations, over- or underestimation and unwarranted conclusions and only can be achieved by having reliable analytical data [18]. The specific GC conditions need to be optimized and verified regularly, although GC is the most extensively used device for fatty-acid analysis. Therefore, the calibration process for GC sample analysis must be implemented [12]. The method must be validated in linearity, precision, sensitivity (limit of detection and limit of quantification), accuracy and specificity. Ensuring that the validated approach is appropriate for its intended use, the analysis should be extended to the matrix used. 

This study was aimed to validate the common method for the omega-3 FA determination using GC-FID in the raw and cooked, high-omega-3, warm-water fish, selayang [19]. The use of a validated method is important to understand the effect of a wet cooking method on the oxidation of EPA and DHA that will be evaluated kinetically using the Weibull model. High retention of EPA and DHA in cooked fish is one of the vital sources of dietary omega-3 fatty acids for the community. Previously, Wan Rosli et al. (2012) found seven marine fish species considered excellent DHA sources with DHA exceeding 100 µg/g, including Indian scad (Selayang), hardtail pomfret, black pomfret, Delagoa threadfin bream, anchovy, spotted mackerel, barramundi and sixbar grouper [19]. Selayang fish is one of the most frequently consumed, abundant and cheap fish in Malaysia [20,21]. Nevertheless, the determination of fatty acids is quite timely and laborious. However, chemical analysis is very costly and impossible to carry out daily. Because of this, for sample preparations, this study used a smaller amount of sample (40 g) and solvents, such as chloroform and methanol, for extraction (60 mL of each per replicate) compared to Bligh and Dyer (1959) [22]. The Bligh and Dyer procedure has been extensively used for extracting lipids from food (more than 34,000 citations according to Web of Knowledge). The procedure is implemented at room temperature to preserve the integrity of lipid material and avoid oxidation processes [23]. To shorten the time for extraction, the centrifuge was then used for mixtures separation. With all these changes, therefore, it is also necessary to validate the methods. The method might be helpful, not only to save time and money, but to produce high recoveries and good precision results for omega-3 fatty-acid determination in the fish. This study hoped to provide reliable data and can be used to support the respective research in the future. 

## 2. Results and Discussions

### 2.1. Method Development

The analytical methods were divided into two types: standard and non-standard. Those validated, developed, peer-reviewed and published by regulatory bodies are categorized as a standard method. In contrast, the non-standard method is either the in-house method or the method based on research publications and application notes from equipment manufacturers [24]. The non-standard method has to be validated and verified, according to clause 7.2.2.1 ISO 17025 [25]. Several techniques are being used for the fat extractions in the samples, such as liquid-liquid extractions (petroleum ether, chloroform and methanol) and Soxhlet extraction. The extraction method in this study was done based on Bligh and Dyer (1957) with slight modifications. This study used a smaller amount (40 g) of sample for fatty-acid extraction than Bligh and Dyer (100 g). The extraction solvents, such as chloroform and methanol, were also smaller: 60 mL of each per replicate, compared to Bligh and Dyer (200 mL of each per replicate). Primarily, the methanolic chloroform (60 mL methanol and 30 mL chloroform) was mixed with the samples, then homogenized using a Stomacher blender for 120 s. The samples were then added with 30 mL chloroform before being homogenized again for 30 s. The homogenate samples were added with 30 mL distilled water then centrifuged for 10 min at 3000 rpm. To save time, the environment and exposure to hazardous chemicals, the filtration step of homogenate was substituted with centrifugation, which shortens the separation time of the organic and aqueous layers to 10 min. As a result, this study successfully separated the EPA and DHA contained in the standard mixture (Figure 1) and fish samples. 

### 2.2. Validation Parameters

#### 2.2.1. Linearity and Sensitivity

Quantitative analysis was performed using 37 FAME Mix (Supelco, Bellefonte, PA, USA). The FA standards were identified by retention times and compared with profiles of chromatograms in the certificate of analysis of the standard mix. They were detected in 45.87 min of GC-FID analysis, with a proper separation analysis between peaks. The linear range was initially tested between 0.156 mg/mL and 5 mg/mL. The calibration curve plotted within a working range at 0.156 and 5.0 mg/mL. Table 1 shows the retention time (Rt), linearity ranges, the equation, correlation coefficients and detection limits (limit of detection (LOD) and quantification (LOQ)) of calibration curves for each FA standard. From the table, all standards appeared to be linear over the concentration range studied and the coefficient of determination; R² were higher than 0.990 for all compounds. This indicated more than 99% of the detector’s signal variance were explained by concentration changes and the correlation efficiency was excellent [26]. Therefore, the calibration model fit well with all compounds. These results also indicated adequate sensitivity of the tested method, in which the sensitivity of the GC-FID, LOD and LOQ were between 0.109 mg/mL and 0.177 mg/mL and 0.332 mg/mL and 0.537 mg/mL, respectively for all the target compounds. 

#### 2.2.2. Precision (Repeatability and Intermediate Precision)

The precision of the method was assessed via repeatability and intermediate precision. Table 2 and Table 3 show the repeatability and average intermediate precision data, respectively. The repeatability was based on nine (*n* = 9) complete analyses (any three concentrations of spiked samples: S1-low (0.156 mg/mL), S2-medium (1.25 mg/mL) and S3-high (5 mg/mL). The cooking treatment fish samples (A (raw), B (steamed) and C (baked in foil)) were spiked with pure methyl esters DHA and EPA standards (Supelco, Bellefonte, USA) and injected triplicates under the same conditions in a day. In contrast, the intermediate precision was established from nine complete analyses (*n* = 3) of every sample for three consecutive days. Table 2 and Table 3 indicate that RSD% ranged from 0.002% to 0.028% for intraday and from 0.012% to 0.092% for interday respectively. The precision was expressed as a percentage of the relative standard deviation of concentrations (RSD%). According to the FDA Reviewer Guidance for the Validation of Chromatographic Methods (1994), the acceptable value for each concentration should be less than 1% [27]. The good precision value indicated no variability in precision at different concentrations measured on the same and different days 

The study result was in line with a study by Oviedo Castrillon et al. (2016) that found repeatability (RSD%) of less than 1%, in which *n*-3 PUFA (C18:3n-3, C20:5n-3 and C22:6n-3) were 0.6%, 0.5% and 0.1%, respectively, in a commercial tuna sample that was prepared in sextuplet and injected in triplicate [28]. In a study by Trbovic et al. (2018), the reproducibility (RSD%) for majority *n*-3 PUFA (C18:3n-3, C20:5n-3 and C22:6n-3) in fish tissues were found to be less than 3%, a bit higher compared to this study [29]. Another study by Abdel-Moemin et al. (2015) showed RSD values lower than 6% and 7% for intraday and interday, respectively, in the fish sample (Nile tilapia) spiking with three levels of fatty-acid (FA) standards (50%, 100% and 150%) that were assayed in triplicate during the analysis [30]. Another study by C. Truzzi et al. (2017) indicated intraday and interday precision for major FA were less than 4% and 7%, respectively, in Antarctic fish muscles that were fortified with FAME standard solutions [26]. Salimon et al. (2017) showed that RSD ranged between 0.89% to 2.34% for repeatability and 1.46% to 3.72% for reproducibility of food containing fat samples [2]. 

#### 2.2.3. Accuracy

For recovery percentage (R%), the three fish samples used for the study (steamed (A), baked (B) and raw (C)) were spiked with the pure methyl esters DHA and EPA standards (C22:6 and C20:5) and analyzed in triplicate. The recovery percentages (difference between spiked concentration and blank sample/expected concentration × 100) obtained for each FA studied is shown in Table 4. For the samples spiked with DHA standard (C22:6), recovery values for high-concentration standard added (5 mg/mL) ranged from 89.64% to 120.03%, whereas for the medium-concentration standard spiked to the sample (1.25 mg/mL) ranged from 89.81% to 96.82%. Meanwhile, the low-concentration DHA standard recovery range added to the samples (0.156 mg/mL DHA) showed good recovery values ranging from 90.58% to 97.02%. Good recoveries also indicated that the derivatization method used to determine fatty acids in fish samples was appropriate. There was a small effect detected when the analytes were added and analyzed on different days. According to Indarty et al. (2003), temperature and heating times could significantly affect total fatty-acid recovery; temperatures of 90 °C and 100 °C and reaction times of 90 min and 30 min could give maximum recovery of TFA extracted from oil [31]. 

The samples spiked with medium and high concentrations of the EPA gave an accepted recovery percentage from 90.26% to 99.67% and 124.25% to 125.78%, respectively. Except for the samples spiked with a low standard concentration, they showed a slightly higher recovery percentage ranging from 148.71% to 156.22%. The effects could be due to the processes between preparation of samples and FA analysis, such as loss due to evaporation or partial degradation, lower reaction temperature or incomplete transesterification of the internal standard [32].

Recovery is the information we could use to correct the results. The recovery values between 80–120% were acceptable and indicated that the matrix did not influence the method. Therefore, the method was considered adequate for successfully quantifying FAME from vegetable oil, animal oil and fat samples [26,32]. Most of the R% in this study were harmonized with a few previous studies. According to C. Truzzi et al. (2017), the mean recovery FA fortified with low and high concentrations of FAME standard were 96 ± 9% (min–max, 81–115%) and 96 ± 7% (min–max, 81–111%), respectively, in an Antarctic fish muscle [26]. Meanwhile, the recovery (R%) of *n*-3 PUFA (C18:3n-3, C20:5n-3 and C22:6n-3) were 93.5%, 116.2% and 105.3%, respectively, as conducted by Oviedo Castrillon et al. (2016) using a commercial tuna oil sample extracted using the Bligh and Dyer method [28]. Most of the R% in this study was comparable to our previous study that indicated satisfactory R% at about 90–120% in extracted fishes using the Bligh and Dyer method spiked with FA standard [33]. 

Overall, the recovery results gave accepted values for almost all the samples. Therefore, the derivatization method used to determine fatty acids in fish samples seemed appropriate. There was only a minimal effect detected when the analytes were added and analyzed on different days. The heat applied during steaming and baking in the foil cooking process did not substantially affect DHA and EPA changes in fish. This result was in line with a few studies that reported that some cooking methods, such as steaming and baking in foil, succeeded in retaining the *n*-3 PUFA percentage in fish compared to other cooking methods [33,34,35].

#### 2.2.4. Specificity and Selectivity

Specificity is the ability to analyze a single compound of interest unequivocally. In contrast, selectivity is the ability to differentiate the interest among other substances or in the presence of interference in the samples [24]. In this study, specificity was determined by comparing the peak with and without compounds of interest. Figure 1 shows no interfering peaks were observed when injecting 1 μL of the diluent (hexane) into the system at the retention time of long-chain PUFA and other fatty acids. In contrast, Figure 2 shows EPA and DHA peaks eluted and analyzed with the same methods. Figure 3 shows the methyl nonadecanoate (C19:0) standard was injected and eluted using the same method.

Table 5 shows the DHA and EPA peak resolutions before and after spiking the sample with methyl nanodecanoate (C19:0). As shown in Table 5, for selectivity, DHA and EPA of samples were not affected by adding methyl nonadecanoate (C19:0) standard into the samples. Good chromatographic separation was demonstrated by Rs > 1.5 for EPA in all samples spiked with C19:0. Acceptable Rs was found (Rs > 1.0) for DHA in almost all samples except for DHA in raw fish (0.849). Above figures showed that the flame ionization detector (FID) was highly selective to the target components. These figures also showed that FID is a widely used detector for GC and sufficient for food analysis using GC [15].

## 3. Materials and Methods

### 3.1. Materials

The method validation study was verified using a selayang fish fillet sample (*Decapterus maruadsi).* It is also known as Indian scad, curut, busung or sardine among Malaysians. The locality of fish was Kuala Selangor but purchased from a local market in Selayang, Kuala Lumpur. The fish was chosen because it is among Malaysia’s most frequently consumed fish, instead of others, such as Indian and short-fin scad, mackerel, anchovy, yellowtail and yellowstripe scad, tuna, sardine, torpedo scad, pomfret, red snapper, and king mackerel [20]. The fish is also very affordable in price and abundant in Malaysia. The fish is also considered an excellent DHA source with a DHA reading exceeding 100 µg/g, including others, such as hardtail pomfret, black pomfret, Delagoa threadfin bream, anchovy, spotted mackerel, barramundi and sixbar grouper [19]. Besides being considered the right choice of DHA, they represent a precious essential nutrient choice for healthy body maintenance.

The fish samples were purchased in sufficient quantity to preserve homogeneity in sampling and minimize variation possibilities. Fishes with the same length range (19–30 cm) and weight (250–400 g) were selected. They were preserved in clean, zipped, polythene bags and transported to the laboratory in an iced-filled polystyrene bag insulation box. Upon arrival, the fish were then beheaded and cleaned multiple times with tap water to remove adhering blood and excessive mucus. The fishes were stored in ice-cold water (0 °C) for 5 min before eviscerating and beheading. Subsequently, the fish samples were filleted, then divided into three groups, each consisting of two fillets. The first group was the uncooked fish that served as the control, whereas the other two groups were cooked using two different temperatures in the following wet cooking methods: baking in foil (160 °C) and steaming (100 °C). Our previous study by Choo et al. (2018) found that yellowstripe scad fillets cooked in a steaming method retain more DHA and EPA than baking in foil, grilling and deep-frying cooking methods [36]. The raw and cooked samples were sealed in polythene bags and kept under cold conditions (−80 °C) until further analysis.

### 3.2. Chemicals and Standards

The solvents used for sample preparations, methanol (MeOH) and chloroform (CH_3_CH_3_), were GC grade (Merck KGaA, Darmstadt, Germany). The 37-component FAME mix 47885-U, single-stock solutions for the target compounds and the long-chain fatty acids standards (DHA and EPA) were purchased from Sigma–Aldrich (Supelco, Bellefonte, USA). Methyl nonadecanoate acid (C19:0, Supelco, Bellefonte, USA) was used as an internal standard. All solvents and reagents for sample preparations were analytical grade and purchased from Sigma–Aldrich.

### 3.3. Fat Extraction and Derivatizations of the Samples

The fat extraction was done according to the Bligh and Dyer method with slight modifications [22]. Representative samples of fish fillets (40 g) were homogenized using a Stomacher blender (Bagmixer 400, Interscience, France) for 2 min with a mixture of methanol (60 mL) and chloroform (30 mL). One volume of chloroform (30 mL) was added to the mixture, and after blending for additional 30 sec, distilled water (30 mL) was added. The homogenate was stirred with a glass rod and centrifuged (3000 rpm, 10 min). The lower clear phase was drained into a 250 mL, round-bottom flask and concentrated with a rotary evaporator at 40 °C. The extracted lipids were kept in a solvent containing 0.05% butylated hydroxytoluene (BHT) in glass bottles, flushed with nitrogen and wrapped with aluminum foil to avoid light exposure and minimize oxidation. The samples were stored immediately at −80 °C and only removed from the freezer for further analysis. 

For the preparation of fatty-acid methyl esters (FAMEs), approximately 25 mg (±0.1 mg) of oil was weighed and added to 1.5 mL of NaOH 0.50 M in methanol in a 15 mL capped centrifuge tube. The mixture was heated in a water bath at 100 °C for 5 min, then cooled at room temperature. Next, 2.0 mL boron trifluoride (BF_3_, 14%) in methanol was added to the mixture and heated again in a water bath at 100 °C for 30 min. The tube was then cooled in running water to room temperature before adding 1 mL of isooctane. Next, the tube was vigorously stirred for 30 s before adding 5.0 mL of a saturated sodium chloride solution to facilitate the phase separation. Next, the esterified sample was placed in a refrigerator and left to rest for better phase separation. After collecting the supernatant, another 1.0 mL of isooctane (containing 0.05% BHT as an antioxidant) was added into the tube and stirred. Finally, the supernatant was collected and added to the previous fraction. The sample was then concentrated to a final volume of 1.0 mL for later injection into the gas chromatograph. As precautions, amber vials were used to minimize oxidation during analysis.

### 3.4. Chromatographic Analysis

A gas chromatography system equipped with a flame ionization detector (GC-FID) and an automated liquid sampler was used to analyze development and validation methods (Agilent Technologies, Santa Clara, CA, USA). A highly polar column, the HP-88 (60 m length, 0.25 mm ID, 0.2 μm DF), was used to separate FAMEs (Agilent, Santa Clara, CA, USA). Helium was used as the carrier gas, with an average velocity of 20 cm/sec. The column flow was 1 mL/min at 19.39 psi. Split injection with a split ratio (volume of gas passing down the capillary column) of 20:1 was used. The operating conditions were programmed at a 250 °C injection port, 250 °C flame ionization detector and 230 °C column temperature. The oven temperature was set at 40 °C for 0.5 min, raised to 195 °C at 25 °C/min (for 25 min), raised to 205 °C at 15 °C/min (for 3 min) and finally raised to 230 °C (for 8 min) at a rate of 10 C/min. The whole analysis took 45.87 min. The optimization of the chromatographic condition was carried out by modifying the flow rate, the column temperature and the time for analysis to get the targeted fatty acids in this fish. Compounds were identified by comparing the retention times of 37-component FAME mix 47885-U (Supelco, Bellefonte, PA, USA). 

### 3.5. GC-FID Method Validation

The whole analysis protocol of the GC-FID method was validated in terms of precision, accuracy, linearity, sensitivity and specificity, as per ICH method validation guidelines [37]. 

#### 3.5.1. Linearity 

Linearity was assessed by injecting serially diluted 37-component FAME standards (CRM47885) into the system. Each fatty-acid stock solution was prepared daily from Std 1 by diluting half a volume with *n*-hexane to obtain five concentration levels (Std 2 to Std 6). The ICH guideline recommended that at least five concentrations be run to establish linearity [37]. Calibration curves were formed for each of the compounds. The data of peak-area response (Y) at each concentration (X) was used to construct a linear regression model. The linearity parameters, linear regression and the coefficient of determination (Rº), were obtained from the linear relationship between the peak area and the concentrations of the fatty-acid standards. The slope, b, intercept, α and the coefficient of determination (Rº) were determined using the Excel lines function on peak-area ratio at different calibration levels. 

#### 3.5.2. Sensitivity 

The method sensitivity was calculated through detection limit (LOD) and quantification limits (LOQ), which were determined from the calibration curves of the FA standard. The LOD was calculated statistically as 3.3 σ/b and the LOD was calculated as 10 σ/b, in which σ is the residual standard deviation of the calibration curve (Sx/y), and b is the slope of the regression line of each compound in the calibration curves.

#### 3.5.3. Precision

The precision of the method was evaluated at two levels: repeatability and intermediate precision. The intraday precision (repeatability) was evaluated using three preparations, each containing two fatty acids, analyzed in triplicates within the same day. The intermediate precision of ruggedness can be expressed by lab variations, such as using different days or different analysts or equipment in the same lab [35]. This study compared data from three successive days using the same method of sample analysis to gain intermediate precision data. Precision is expressed by the relative standard (*RSD*%) of the retention times and peak areas and calculated as below:(1)$$RSD\%=\frac{s}{x}\times 100$$
where *s* is a standard deviation of replicates and *x* is the measurements’ mean. The acceptable value for repeatability is between ½ and two times the calculated values [21].

#### 3.5.4. Accuracy

Accuracy of a measurement result showed the closeness of results to the actual value and verified utilizing recovery assay. The accuracy was obtained by spiking known amounts of standards (EPA and DHA) with three concentration levels (0.156 mg/mL, 1.25 mg/mL and 5.0 mg/mL) into the fish samples (within the working range) and analyzing in triplicate before and after spiking. The variation inaccuracies were then calculated from the response of samples that spiked with known amounts of analyte against standards on FID detection to ensure no interference existed. The recovery percentages were calculated as follows and were reported as recovery percentage (*R*%):(2)$$R(\%)=\frac{C_{a}-C_{b}}{C_{f}}\times 100$$
where *C_a_* is the concentration of the spiked samples, *C_b_* is the concentration of the sample before spiking and *Cf* is the true added concentration.

#### 3.5.5. Selectivity and Specificity

Selectivity is the ability to unequivocally assess the target analytes in the presence of other analytes or interference in the samples [24]. This method was tested in the presence of another analyte that was most likely not in the sample. The selectivity was determined by the following equation:(3)$$Rs=2\times \frac{\left(t_{2}-t_{1}\right)}{\left(W_{1}+W_{2}\right)}$$
where *Rs* is resolution, expressed as retention times (min) of the two peaks, *t*_2_ and *t*_1,_ and the baseline widths, *W*_1_ + *W*_2_. Specificity is the ability of the method to measure an analyte in the presence of interference. The retention times of related fatty acids were confirmed by comparing the retention times with those obtained from each injected individual fatty acid. 

### 3.6. Statistical Analysis

During method validation and development, the means, standard deviations, coefficient variation (CV) and RSD% were calculated using Microsoft^®^ Excel Professional Edition 2010 (Microsoft Corporation, Redmond, WA, USA). The correlation and regression analyses were performed using SPSS version 17.0 (Chicago, IL). The value *p* < 0.05 was taken as statistically significant. One-way analysis of variance (ANOVA) was used for the determination of significance of the process in method development and validation.

## 4. Conclusions

A method with good precision (RSD ≤ 2%), high accuracy (R > 95%) and selectivity to determine and quantify EPA and DHA in the fish sample was developed in this study. A high relationship between the FA concentration in the sample and the chromatograms response area was revealed. In conclusion, this method can be proposed to quantify DHA and EPA simultaneously and the 37 fatty acids in general. 

## Figures and Tables

**Figure 1 molecules-26-06592-f001:**
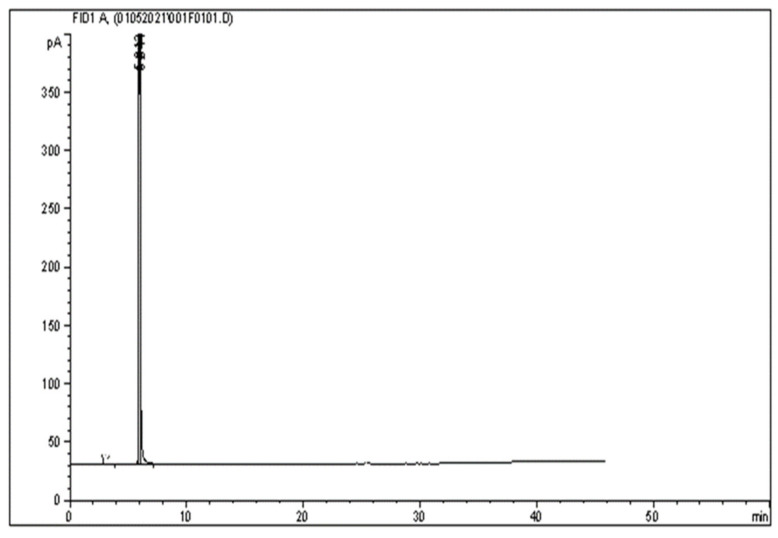
GC chromatogram of 1 μL of diluent (hexane) injected into the system.

**Figure 2 molecules-26-06592-f002:**
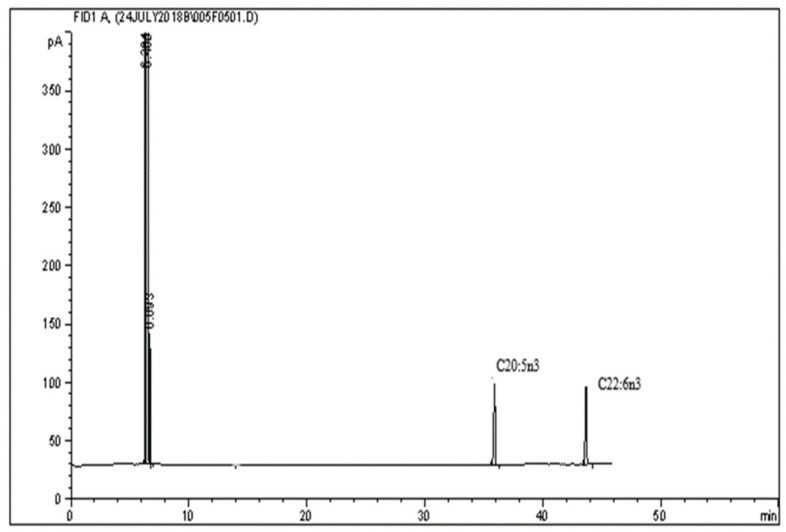
GC chromatogram of 1 μL of 5 mg/mL of EPA and DHA standards injected into the system.

**Figure 3 molecules-26-06592-f003:**
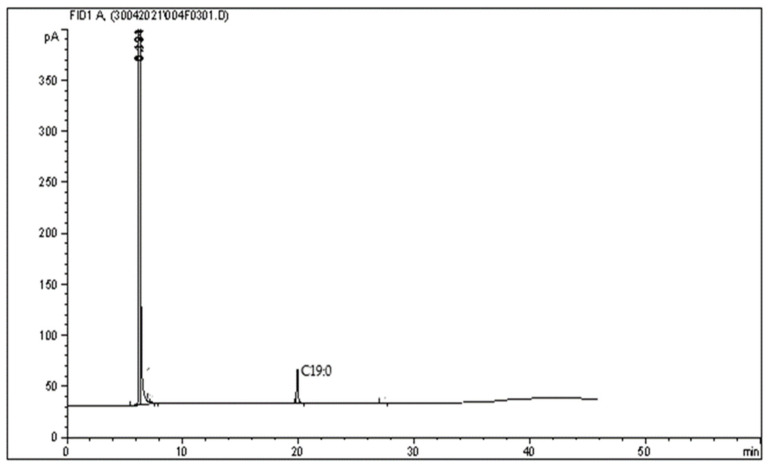
GC chromatogram of 1 μL of 2.5 mg/mL methyl nonadecanoate (C19:0) standard analyzed and injected using GC.

**Table 1 molecules-26-06592-t001:** Linearity and estimated regression parameters for FA standard obtained from calibration curves.

Fatty Acids	Retention Time (*R*_t_)	Linear Ranges (Std ^1^ 2-Std6) (mg/mL)	Calibration Curve Equation	R ^2^	LOD ^3^ (mg/mL)	LOQ ^4^ (mg/mL)
C4	6.573	4.66–0.231	y= 0.0004× + 0.0043	0.9907	0.177	0.537
C6	8.143	5.066–0.286	y= 0.0057× − 0.001	0.9956	0.132	0.401
C8	9.533	5.026–0.272	y = 0.0069× − 0.0009	0.9969	0.109	0.332
C10	10.692	5.023–0.279	y = 0.0077× − 0.0011	0.9965	0.116	0.352
C11	11.728	5.205–0.326	y = 0.008× − 0.001	0.9967	0.114	0.344
C12	12.246	5.066–0.288	y = 0.004× − 0.0005	0.9964	0.119	0.362
C13	12.791	5.019–0.271	y = 0.0083× − 0.0011	0.9965	0.116	0.353
C14	13.386	5.019–0.276	y = 0.0041× − 0.0005	0.9961	0.123	0.373
C14:1	14.062	5.079–0.289	y = 0.0083× − 0.0012	0.9962	0.124	0.375
C15	14.705	5.041–0.289	y = 0.0041× − 0.0006	0.9961	0.124	0.377
C15:1	14.852	5.112–0.30	y = 0.0042× − 0.0006	0.9954	0.136	0.414
C16	15.628	5.015–0.275	y = 0.004× − 0.0005	0.9958	0.128	0.387
C16:1	15.799	5.053–0.283	y = 0.0127× − 0.0019	0.9958	0.129	0.392
C17	16.587	5.109–0.289	y = 0.0042× − 0.0006	0.996	0.128	0.387
C17:1	16.956	5.110–0.289	y = 0.0036× − 0.0005	0.9954	0.149	0.452
C18	17.928	5.016–0.282	y = 0.0043× − 0.0006	0.9956	0.130	0.395
C18:1n9t	18.397	5.066–0.352	y = 0.0086× − 0.0011	0.9934	0.137	0.415
C18:1n9c	19.072	5.273–0.281	y = 0.0043× − 0.0006	0.9955	0.138	0.408
C18:2n6t	19.438	5.052–0.292	y = 0.0086x − 0.0013	0.9953	0.136	0.412
C18:2n6c	20.239	5.035–0.350	y = 0.0038× − 0.0004	0.9933	0.136	0.413
C20	21.203	5.003–0.289	y = 0.0041× − 0.0006	0.9955	0.132	0.399
C18:3n6	22.497	5.082–0.364	y = 0.0086× − 0.0012	0.9929	0.142	0.430
C20:1	22.698	5.088–0.281	y = 0.0039× − 0.0005	0.9958	0.130	0.393
C18:3n3	23.707	5.089–0.288	y = 0.0039× − 0.0005	0.9953	0.137	0.416
C21	24.071	5.011–0.289	y = 0.0044× − 0.0006	0.9953	0.135	0.410
C20:2	25.423	5.047–0.312	y = 0.0045× − 0.0007	0.9949	0.142	0.43
C22	26.832	5.017–0.300	y = 0.0041× − 0.0006	0.9949	0.140	0.425
C20:3n3	29.164	5.058–0.293	y = 0.012× − 0.0019	0.9952	0.138	0.419
C22:1	30.777	5.030–0.285	y = 0.0036× − 0.0005	0.995	0.140	0.423
C20:3n6	31.067	5.024–0.294	y = 0.0035× − 0.0005	0.9953	0.135	0.410
C23	31.582	5.102–0.308	y = 0.0042× − 0.0006	0.9941	0.154	0.468
C20:4	33.404	5.997–0.286	y = 0.0046× − 0.0007	0.9949	0.141	0.427
C22:2	34.904	5.035–0.303	y = 0.004× − 0.0006	0.9953	0.137	0.413
C24	35.201	5.121–0.307	y = 0.0034× − 0.0005	0.9952	0.139	0.421
C20:5n3	37.21	5.025–0.298	y = 0.009× − 0.0016	0.9948	0.142	0.431
C24:1	38.716	4.988–0.298	y = 0.0044× − 0.0003	0.9968	0.111	0.337
C22:6n3	43.114	5.044–0.310	y = 0.0032× − 0.0005	0.9951	0.139	0.421

^1^ Std: standard. R ^2^: coefficient of determination. ^3^ LOD: limit of detection. ^4^ LOQ: limit of quantification.

**Table 2 molecules-26-06592-t002:** Precision (RSD%) under the repeatable condition of spiking 50 uL concentration of EPA and DHA standards in three types of cooked fish intraday.

FAME Std ^1^	RSD ^2^ %								
	Sample A			Sample B			Sample C		
	S1	S2	S3	S1	S2	S3	S 1	S2	S3
C20:5n-3	0.022	0.028	0.003	0.007	0.002	0.012	0.064	0.015	0.009
C22:6n-3	0.012	0.014	0.012	0.005	0.007	0.009	0.026	0.015	0.007

^1^ Std: standard. ^2^ RSD: relative standard deviation.

**Table 3 molecules-26-06592-t003:** The mean intermediate precision (RSD%) of the FA determined in fish samples.

FAME Std ^1^	Day (*n* = 3, mean RSD ^2^ %)		
	Std 1	Std 2	Std 3
C20:5n-3	0.012	0.026	0.078
C22:6n-3	0.023	0.019	0.092

^1^ Std: standard. ^2^ RSD: relative standard deviation.

**Table 4 molecules-26-06592-t004:** Recovery factor (R%) at three additional levels for the three studied samples.

Spiked Std ^1^ Concentration	Sample	DHA (R%)				EPA (R%)			
		1	2	3	Mean	1	2	3	Mean
Low	A	97.02	90.58	94.6	94.06	148.71	156.22	156.05	153.66
Medium	B	96.82	94.07	89.81	93.57	90.26	94.85	99.67	94.93
High	C	117.13	120.03	89.64	108.94	125.78	124.26	124.25	124.76

^1^ Std: standard.

**Table 5 molecules-26-06592-t005:** The peak resolutions (Rs) of the DHA and EPA before and after spiking with methyl nonadecanoate standard (C19:0) in fish samples.

Fatty Acids	Difference in Retention Time (Rt)	Difference in % Area	Rs ^1^
**DHA**			
Raw Fish	0.038	0.22763	0.849
Baked Fish	0.05	0.15458	1.463
Steamed Fish	0.056	0.15515	1.576
**EPA**			
Raw Fish	0.044	0.04963	3.815
Baked Fish	0.068	0.03142	9.237
Steamed Fish	0.081	0.03602	6.987

^1^ Rs: peak resolution.

## Data Availability

The data presented in this study are available in this article.

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
