# Peer review of "Method Development and Validation for Omega-3 Fatty Acids (DHA and EPA) in Fish Using Gas Chromatography with Flame Ionization Detection (GC-FID)"

_molecules, 2021, doi:10.3390/molecules26216592_

Round 1

Reviewer 1 Report

In this article, GC-FID was used as the quantitation method for fatty acids in fish and the method was validated in linearity, precision, accuracy, specificity and sensitivity based on ICH requirement. It is a topic of interest to the researchers in the related areas but the paper needs very significant improvement before acceptance for publication. My detailed comments are as follows:

  1. The results and discussion part are confusing and can’t correspond to the method part. It is recommended that the authors adjust the order of the validation results and clarify the logic of the full text.
  2. The display of the experimental results can be more diversified, and the figures will be more intuitive than tables and numbers.
  3. Please further explain the innovation of this article.

Author Response

Dear reviewer,

Reviewer 2 Report

The authors have presented an interesting subject with an environmental interest in the field of food quality. The GC analysis of fatty acids as FAMEs is a useful tool in the characterization of fats in the determination of total fat and trans-fat content in foods. The manuscript has been well prepared, based on recent literature. However, I have found many papers in this field, so it seems that the idea is not new, and this paper does not impressive deal with this subject. The article has scientific soundness and introduces significant merit, but the objective of this work is not clearly formulated. However, there are numerous less or more important aspects that authors should improve.

  • In general, some parts of the manuscript are difficult to read with methodological, stylistically and logical errors.
  • The validation process is an essential element of the analytical method. I am not convinced that the validation procedure itself is a novelty in the work without an innovative approach to this process.
  • The recovery was not estimated properly.
  • In my opinion, the sensitivity was not correctly estimated, and this section should be described in detail.
  • The linearity ranges should be changed and should include limits of quantification.
  • The selectivity of the method should also be estimated.
  • There are many different columns to perform FAME analysis by the GC method. The column HP-88 is not the best choice for this purpose.
  • In my opinion, it should be better justified why the GC-FID method was chosen?
  • Showing the obtained chromatograms will be appreciated.
  • Some of the suggestions are listed below:

The title: should be rewritten (Development and validation …?).

Line 13: Gas chromatography with flame ionization detection looks better.

Line 35-36: The sentence sound laconic and should be corrected.

Line 106: The correlation coefficients (r) is not the same as the r2 - coefficient of determination.

Line 118: Comparing quantification limits for molecules with different molecular mass using mass units does not seem to make sense.

In my opinion, the manuscript requires major revision before any other consideration and does not meet with criteria of the journal.

Author Response

Dear Reviewer,

Round 2

Reviewer 2 Report

I would like to thank the Authors for considering my suggestions, but some aspects still need to be improved:

  • Title: Please correct to “…gas chromatography with flame ionization detection”
  • The novelty of the work is based on the sample pretreatment, not on the chromatographic technique used.
  • Table 1 should be corrected, especially positions C17 and C17:1. The linearity ranges should be changed and should include limits of quantification. Please note that the LOQs are outside the linear ranges.
  • Column Dead time on the presented chromatogram seems to be quite long and suggest some problem with column flow. The linear velocity of 20.0 mL/min is too high in my opinion and normally should be presented as cm/s. Please verify that the column flow rate and pressure are correct. recommended flow rate should be set up to 1,8 mL/min for this column type and carrier.
  • 1ul should be corrected to 1 μL
